ⓐ | **Open Peer Review** | Genetics and Molecular Biology | Research Article

# Characterization of biofilm formation by *Exiguobacterium* strains in response to arsenic exposure

Valentina B. Pavez,[1] Nicolás Pacheco,[1] Juan Castro-Severyn,[2] Coral Pardo-Esté,[3] Javiera Álvarez,[1,4] Phillippi Zepeda,[1] Gabriel Krüger,[1] Karem Gallardo,[5,6] Francisco Melo,[7] Rolando Vernal,[8] Carlos Aranda,[9] Francisco Remonsellez,[2,5] Claudia P. Saavedra[1]

**ABSTRACT** The Salar de Huasco (SH) salt lake in northern Chile is an extreme environment characterized by high atmospheric pressure, UV radiation, salinity, variable temperatures, and the presence of heavy metals, including arsenic. *Exiguobacterium* bacteria have adapted to thrive in these challenging conditions and possess various resistance mechanisms, including biofilm formation, redox reactions, methylation, and altered respiration. In this study, *Exiguobacterium* strains isolated from the SH were assessed for their capacity to form biofilms in the presence of arsenic, a metalloid that exists in different oxidation states, in order to understand their resistance mechanisms to this heavy metal. The minimum inhibitory concentration (MIC) of each strain against different concentrations of arsenic [III] and [V], biofilm formation using crystal violet staining, and the expression of genes related to biofilm formation were evaluated. The structure of the biofilms was characterized using scanning electron microscopy (SEM) and atomic force microscopy (AFM). Furthermore, the extracellular polymeric substances (EPS) produced during biofilm formation were purified, quantified, and their composition determined. The results showed that the tested *Exiguobacterium* strains exhibit a significant ability to form biofilms when exposed to arsenic. This biofilm contributes to their arsenic resistance, shedding light on the underlying mechanisms. These findings enhanced our understanding of the biofilm formation process, its role in arsenic resistance, and the adaptive strategies employed by bacteria in extreme environments. This study also contributes to the field of microbial resistance mechanisms that have implications for environmental and biotechnological applications.

**IMPORTANCE** In this work, we characterized the composition, structure, and functional potential for biofilm formation of *Exiguobacterium* strains isolated from the Salar de Huasco in Chile in the presence of arsenic, an abundant metalloid in the Salar that exists in different oxidation states. Our results showed that the *Exiguobacterium* strains tested exhibit a significant capacity to form biofilms when exposed to arsenic, which would contribute to their resistance to the metalloid. The results highlight the importance of biofilm formation and the presence of specific resistance mechanisms in the ability of microorganisms to survive and thrive under adverse conditions.

**KEYWORDS** *Exiguobacterium*, biofilm, arsenic, resistance, poly-extremophilic, atomic force microscopy (AFM), extracellular polymeric substances (EPS)

Arsenic (As) is recognized as one of the major environmental hazards related to heavy metal pollution (1). The presence of As in the biosphere is mainly due to volcanic events and anthropogenic activities related to the processing of geological materials such as coal and minerals and the use of biocides by agricultural and forestry industries (2, 3). Various toxicological studies have revealed that prolonged exposure to As triggers a myriad of harmful effects for human health, such as the development of specific

Address correspondence to Claudia P. Saavedra, csaavedra@unab.cl.

The authors declare no conflict of interest.

See the funding table on p. 17.

types of cancer, and reproductive, pulmonary, and neurological diseases (4). It naturally concentrates in water and soils and is systematically accumulated in products consumed by human, when untreated soils, groundwater, and streams are used in crop cultivation and processing (5).

The World Health Organization recommends an upper limit of 10 µg As/L in water for human consumption. However, concentrations in the order of milligrams of As/L have been detected in water intended for consumption, significantly exceeding the prescribed limit (6). Considering this scenario, it is of utmost importance to develop methods and technologies capable of permanently and effectively removing As from soil and water. Currently, chemical and physical methods are employed to remove As from the environment, such as filtration, osmosis and reverse electrodialysis, nanofiltration, and coagulation (7).

The use of microorganisms as a means of bioremediation has gained strength as an alternative to conventional environmental decontamination processes. The detection of microorganisms with essential genes that encode phenotypes responsible for metal-bacteria interactions and the formation of biofilms are crucial steps for the design of successful bioremediation strategies for metals and radionuclides (8, 9). Biofilm formation consists of three fundamental steps, namely, adhesion, growth, and detachment or release. In the first step, the bacteria perceive the surface and proceed to form a union through the flagellum or pili. In the second step, once the bacteria adhere, growth is observed by cell division, thus forming microcolonies at the union site. As the cells continue to colonize the surface, the secretion of specific molecules that contribute to the stabilization and robustness of the biofilm is triggered, such as extracellular polymeric substances (EPS), which constitute a large part of the biofilm matrix along with proteins, nucleic acids, and other polymers responsible for the three-dimensional structure of the biofilm. Finally, when the biofilm reaches maturity, some bacteria are released from the matrix to colonize new surfaces, thus ending the process of biofilm formation and development (10).

Previous reports have described the importance of combining tolerant/resistant bacterial strains in biofilm matrices for bioremediation, referring to the set of bacterial cells incorporated in an EPS matrix (11). The EPS secreted by the biofilm favor immobilization, sorption, sequestration, and precipitation of heavy metals (12). On the other hand, microbial communities congregated in a biofilm are more stable and can survive for longer periods when exposed to harsh environmental pressures, such as high concentrations of metals and other toxic compounds, salinity, and low water availability. These communities are called polyextremophiles, due to their great capacity to tolerate such unfavorable ecological niches (13). Polyextremophilic microorganisms can be found in the three domains of life, namely, Bacteria, Archaea, and Eukarya, and they can develop in hostile environments where they must resist biotic and abiotic stress conditions (14).

The bacterial genus *exiguobacterium*, which was initially described by collins 1986 (15), is composed of gram-positive, orange-pigmented, halotolerant, facultative anaerobic, non-pathogenic, and motile bacilli (16). This genus has great plasticity, with a capacity to adapt to extreme environmental factors, such as resistance to heavy metals like as. three strains of this bacterial genus, namely, SH31, SH1S21, and SH0S7, have been isolated from the highlands (altiplano) of chile, particularly from the salar de huasco (sh), a place with high concentrations of as (17) (fig. 1).

There are different mechanisms through which different bacteria tolerate As, including reduction, oxidation, respiration, and methylation (19). However, in the genus *Exiguobacterium,* specifically for strains SH31, SH1S21, and SH0S7, the most frequent mechanism discovered is the reduction of arsenate [As(V)] to arsenite [As(III)] and its subsequent expulsion from the cell through ArsB and/or Acr3 pumps. Arsenate enters the bacteria through specific phosphate protein transporters (Pst), while arsenite enters via glycerol transporters (GlpF). In recent studies on the genomic variation in *Exiguobacterium* strains, it was observed that Acr3 is present only in SH31 and SH0S7 strains and is

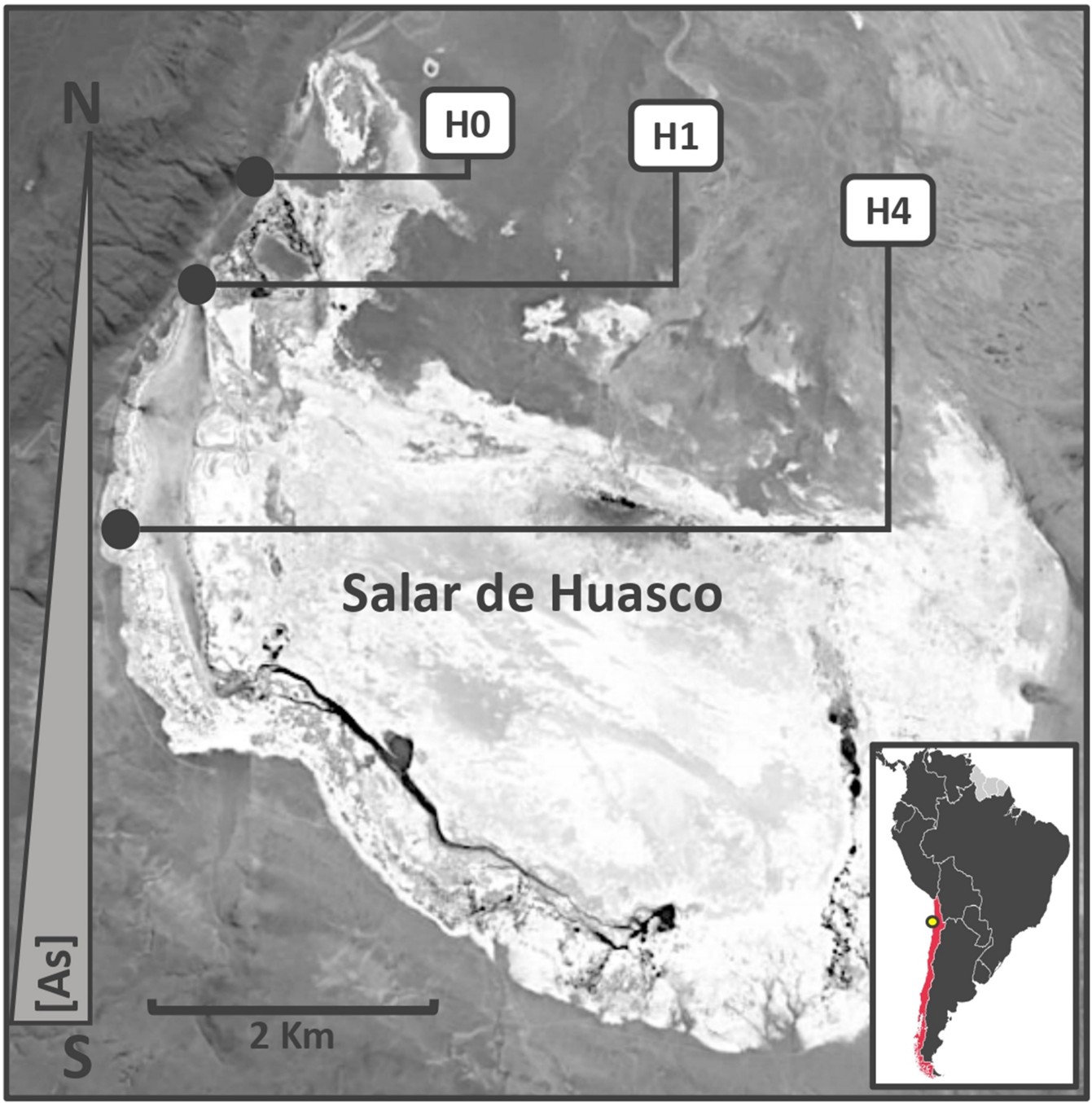

**FIG 1** Sampling site of the *Exiguobacterium* strains SH31 (H4), SH1S21 (H1), and SH0S7 (H0), in the Salar de Huasco, northern Chile. The yellow dot corresponds to the geographic location of the Salar de Huasco. (Modified from reference 18.)

not found in SH1S21, with ArsB being the only expulsion pump described so far for this latter strain (20, 21).

In addition to the Acr3 system, there are other mechanisms responsible for arsenic detoxification/resistance in bacteria. The ArsA protein [ATPase of the As(III) expulsion pump] was identified, which actively participates in the expulsion of As(III), while the arsenite reductase protein (ArsC) plays an important role in the presence of As(V) as it is responsible for the reduction of As(V) to As(III) and could eventually be expressed in the presence of oxidative stress. Other proteins that could be involved in *Exiguobacterium* resistance to As include Cdr, restoring thiols depleted during the reduction process, as

well as ArsK and ArsP in As expulsion processes, as both remove inorganic As from the cell. However, an absolute metabolic model for As treatment within and outside *Exiguobacterium* has not yet been described, and it remains to be elucidated whether As resistance mediated by biofilm formation allows the bacteria to respond and express key genes for maintaining the biological functions that confer As resistance to *Exiguobacterium* (17).

In this follow-up study, the As tolerance profiles and biofilm production of the tested *Exiguobacterium* strains (SH31, SH1S21, SH0S7) were characterized, and the effects of dosage and oxidative states of As on the quality and quantity of biofilm production were determined through quantification and morphological characterization assays.

## MATERIALS AND METHODS

### Bacterial strains

The *Exiguobacterium* strains used in this work were isolated from different points of the Salar de Huasco as described before (17, 22) (Table 1). The following table summarizes the site and conditions of each location (Fig. 1).

### Minimum inhibitory concentration (MIC)

MIC assays were performed to determine the growth capacity under different concentrations of As(III) and As(V). For this purpose, liquid cultures of *Exiguobacterium* were diluted to OD600 = 0.4 in LB + NaCl medium [Add 25 g/L sodium chloride (NaCl), then 15 g/L tryptone, 5 g/L yeast extract, 1 L distilled water, finally adjust to pH 7.4 and autoclave] by adding liquid medium supplemented with different concentrations of arsenic species As(III) ($NaAsO_2$; 0.1–25 mM) and As(V) ($Na_2HAsO_4$; 10–300 mM) in a 1:20 ratio, taking an aliquot of 190 µL of inoculum and adding 10 µL of As(III). In the case of As(V), a volume of 30 µL was taken along with 170 µL of inoculum, thus having a final volume of 200 µL. The samples were then incubated at 37°C for 24 h, and OD600 was measured using a Tecan Infinite 200 PRO nanoquant microplate reader (22).

### Effect of As on biofilm formation

Biofilm formation was quantified in 96-well plates in the presence and absence of As using the crystal violet (CV) staining method. Prior to this, cultures were diluted in a 1:20 ratio by adding the same amount of inoculum used for the MIC assays, with the difference that sublethal concentrations of As(III) and As(V) were used. The plates were incubated for 16, 24, and 48 h at 37°C. After the respective incubation times, the samples were removed with a vacuum pump, washed 3 times with 1% PBS, and then incubated for 1 h at 60°C to fix the biofilm. Subsequently, 1 mL of 0.1% CV was added and left for 15 min. The CV was then removed with a vacuum pump, washed with distilled water, removed again with the vacuum pump, and finally solubilized with 30% acetic acid for 20 min. The extracted CV was measured at an absorbance of 570 nm using a Tecan Infinite 200 PRO nanoquant microplate reader.

### Gene expression associated with biofilm formation

The relative levels of expression of genes directly and indirectly involved in biofilm production were quantified using qRT-PCR. The selected strains were cultured under the

**TABLE 1** *Exiguobacterium* sp. strains used in this study[a]

| Strain | Isolation site | Temperature | Salinity | [Arsenic] | Reference |
|---|---|---|---|---|---|
| SH1S21 | H1 | 14°C | 8.10% | 16.6 mg kg[-1] | (17) |
| SH31 | H4 | 18°C | 77.2% | 155 mg kg[-1] | (22) |
| SH0S7 | H0 | 14.6°C | 11.9% | 9 mg kg[-1] | (17) |

[a]The table shows the isolation site of each strain and the environmental conditions of the substrate at each sampling point (temperature, salinity, and arsenic concentration).

same conditions mentioned above [control without arsenic, half of the specific MIC for both As(III) and As(V)]. Then, the cultures were sedimented, and RNA extractions were carried out using the ZymoBIOMICS DNA/RNA Miniprep Kit, according to the manufacturer's instructions. RNA integrity, quality, and quantity were verified using 1.2% agarose gel electrophoresis, $A_{260/280}$ ratio, and the RNA QuantiFluor System (Promega). cDNA was synthesized using the M-MLV reverse transcriptase kit (Promega) and random primer oligonucleotides (Invitrogen). The qRT-PCR was carried out as follows: 10 min at 95℃ followed by 40 amplification cycles (95℃ × 30 s, 58℃ × 30 s, 72℃ × 30 s), and a final step of 95℃ × 15 s, 40℃ × 1 s, 70℃ × 15 s, and 95℃ × 1 s, using specific primers for each gene (Table 2). Transcription levels were quantified using the Brilliant II SYBR Green qPCR Master Mix (Agilent Technologies) in an AriaMx Real-Time PCR System (Agilent Technologies). Gene expression levels were calculated according to references (23) using the 16S rRNA gene as a normalizer.

## Scanning electron microscopy (SEM)

In order to observe the structure of the biofilm, SEM was performed. Bacterial cultures were added to a 24-well polystyrene plate containing sterile cover slips (13 nm). A 1:20 dilution was added to each well with their respective sublethal concentrations of As(III) and As(V), to a final volume of 1 mL per well. Afterward, cultures were incubated for 48 h, before the removal of the content of each well with a vacuum pump. The cover slips were then carefully washed with 1% filtered PBS. To fix the sample, 2.5% glutaraldehyde was added and incubated overnight at 4℃. Once the samples were fixed, they were analyzed by a JEOL SEM (model JSM-IT300).

## Atomic force microscopy (AFM)

Changes in the surface topography of the bacterial surface induced by As were observed using AFM. Bacterial cultures were added to a 24-well polystyrene plate containing sterile cover slips (13 nm). A 1:20 dilution was added to each well with their respective sublethal concentrations of As(III) and As(V), to a final volume of 1 mL per well. Afterward, cultures were incubated for 48 h, before the removal of the content of each well with a vacuum pump. The cover slips were then carefully washed with 1% filtered PBS. To fix the sample, methanol was added and incubated at 50℃ for 1 h. Once fixed, samples were washed with filtered distilled water and incubated at 50℃ for no more than 15 min. Biofilms were then analyzed by AFM (Nanoscope IIIA by veeco), using the tapping mode at 34 kHz with a silicon nitride cantilever.

## Extraction, quantification, and characterization of EPS

About 10 mL of *Exiguobacterium* cultures under biofilm formation conditions was centrifuged at 15,000 *g* for 2 h at 4℃, and the supernatant was pressure-filtered

**TABLE 2** Primers used in gene expression assays

| Primer | Sequence 5'-->3' | Tm |
|---|---|---|
| dnaK_RT_F | CTC GGA GAT GGC GTG TTT GAA GTT G | 60.1℃ |
| dnaK_RT_R | CTT CGC TTT CTC GGC TGC ATC TTT C | 60.3℃ |
| ywqC_RT F | AGC AAT GAG CTC GGT CGG ATT C | 59.5℃ |
| ywqC_RT_R | CTC ATC GAC TGC GAC CTT CGT AAA | 58.7℃ |
| bdlA_RT_F | TCA AGA TCG CCT CGG ACA TCA A | 59.0℃ |
| bdlA_RT_R | GTT GCC GAG CGT CTT CAA GTT T | 58.4℃ |
| luxS_RT_F | AAG GGA CGG CCG ATA CGA TCT A | 59.6℃ |
| luxS_RT_R | CTC CAT CGT TCA ACA TCG TCC AGT | 58.9℃ |
| fliG_RT_F | CTC ACG TGC GAT TCA ACG GAT T | 57.9℃ |
| fliG_RT_R | ACG ATG CGG CTT TGT GCT T | 58.4℃ |
| 16 s_RT_F | TGC CCC TTA TGA GTT GGG CTA CA | 60.3℃ |
| 16 s_RT_R | TGT GTA CAA GAC CCG GGA ACG TAT | 59.8℃ |

(0.22 µm). Next, the EPS were precipitated from the filtered supernatant using three volumes of cold absolute ethanol (30 mL), and the mixture was incubated at −20°C for 24 h. The precipitate was washed twice with concentrated ethanol and separated by centrifugation. It was then resuspended in Milli-Q water and purified by dialysis against distilled water at 4°C. Excess water was removed by vacuum and lyophilized (frozen samples) for 12 h. The EPS were quantified by weight after lyophilization (total EPS). Total neutral carbohydrates were measured using the phenol-sulfuric acid method (24), in which 100 µg of purified EPS was mixed with 125 µL of concentrated sulfuric acid; 25 µL of phenol (10%) was added to the mixture, vortexed for 1 min, and further incubated at 95°C for 5 min. After cooling, the mixture was placed into wells of 96-well plates, and the absorbance (490 nm) was recorded with a spectrophotometer (Synergy HT, BioTek, USA). Concentrations were obtained with glucose standards. Protein content was determined by the Bradford method with bovine serum albumin as standard. The DNA content in the EPS was obtained by absorbance (260 nm) recorded with a spectrophotometer.

## Attenuated total reflectance-Fourier-Transform InfraRed (ATR-FTIR) spectroscopy

To determine if the surface chemistry of the *Exiguobacterium* cells changes due to As(III) and As(V) exposure, ATR-FTIR spectroscopy (Perkin Elmer Spectrum 100 spectrometer) was used to identify the functional groups on the cell surface. For this, 10 mL of the *Exiguobacterium* cultures under biofilm formation conditions was centrifuged at 15,000 $g$ for 15 min at 4°C, the supernatant was discarded, and the cellular pellets were loaded on the ATR and analyzed directly for their active functional groups in the wavelength range of 700–4,000 $cm^{-1}$.

## RESULTS

### Minimum inhibitory concentration for *Exiguobacterium* strains SH31, SH1S21, and SH0S7 in response to arsenic

In order to determine the resistance capacity of each strain to As, the MIC of the toxicant was calculated. Each strain presents a different resistance range to this metalloid (Table 3). Of note is that As(III) is more toxic, as the data show that the bacteria are more sensitive to this species compared to their capacity to resist higher concentrations of As(V).

### Effect of arsenic on biofilm formation in *Exiguobacterium* strains SH31, SH1S21, and SH0S7

The ability of each strain to produce a biofilm in the presence and absence of As(III) and As(V) was determined after 48 h of exposure to the metalloid (Fig. 2). The greatest formation of biofilms occurred in the presence of As(V) and in the untreated strains, whereas although biofilms were still produced in As(III), their formation was significantly lower. Therefore, these results suggest that for these strains, the production of biofilms contributes toward the resistance against As.

### Gene expression associated with biofilm formation

To better understand biofilm formation, the relative expression of five genes related to this process (stress response, motility, formation) was measured. In general, an induction

**TABLE 3** Minimum inhibitory concentration of the *Exiguobacterium* strains SH31, SH1S21, and SH0S7 in response to As(III) and As(V)

| Strain | As(III) | As(V) |
|--------|---------|-------|
| SH31 | 10 mM | 100 mM |
| SH1S21 | 1 mM | 150 mM |
| SH0S7 | 20 mM | 150 mM |

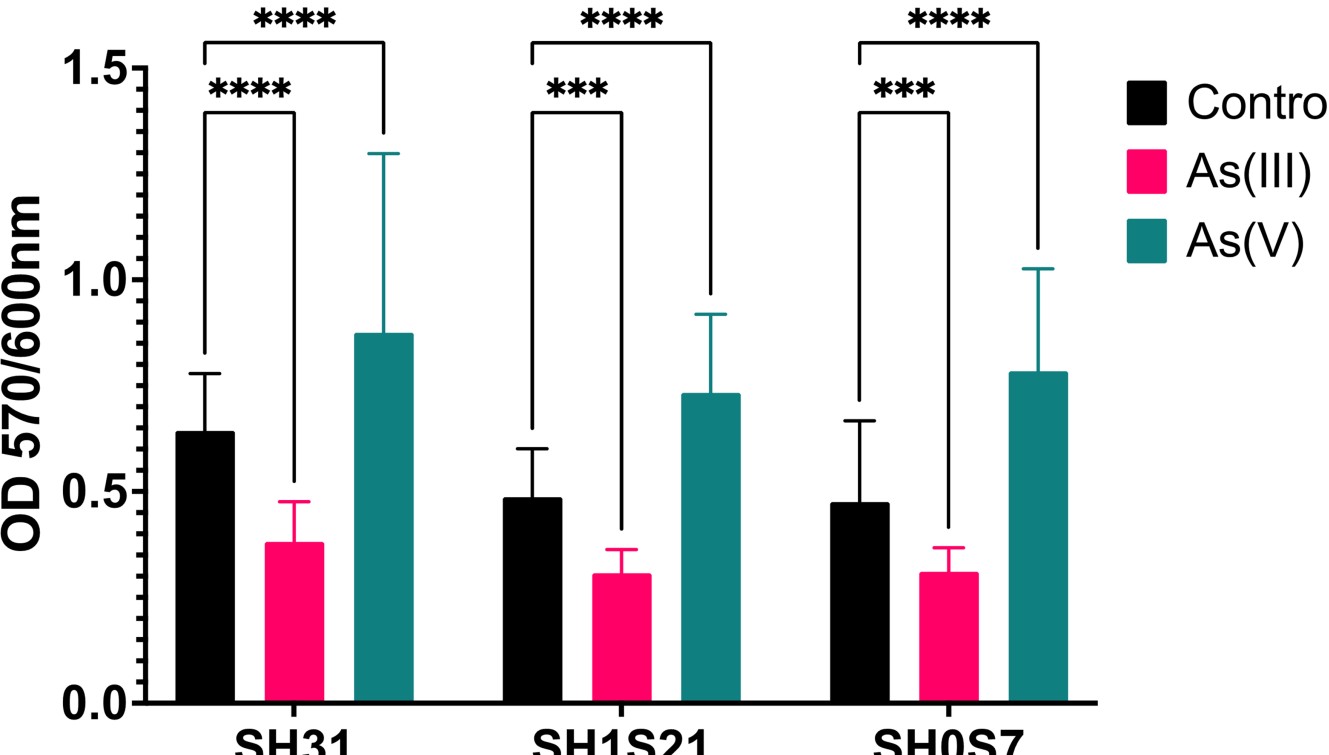

**FIG 2** Biofilm production by *Exiguobacterium* strains SH31, SH1S21, and SH0S7. The black bars correspond to biofilm formation by the strains in the absence of the metalloid, the pink bars correspond to biofilm formation in As(III), and the sea green bars correspond to biofilm formation in As(V). The concentrations used for each strain were 5 mM As(III) and 50 mM As(V) for SH31, 0.5 mM As(III) and 75 mM As(V) for SH1S21, and 10 mM As(III) and 75 mM As(V) for SH0S7. Statistical analysis was performed by two-way ANOVA (ns: not significant, ***$p < 0.001$, ****$p < 0.0001$). Bars represent means ($n = 36 \pm$ SD).

of all evaluated genes was observed in response to As(III) and As(V) regarding control condition (Fig. 3), whereas no significant differences were detected in most cases when comparing the patterns between As(V) and As(III). Of the mentioned genes, *fliG,* which codes for a protein that favors bacterial motility (25), displayed the lowest induction for all three strains (SH31, SH1S21, and SH31). On the other hand, *dnaK*, which plays an important role in the folding of proteins necessary for biofilm formation (26), shows higher expression for the strains SH31 and SH1S21 in response to As(V) than to As(III), which is opposed to what was observed in the SH0S7 strain. Moreover, *bdlA* and *luxS* were the only two other genes that presented significant differences, only in the SH31 strain and in both cases with a greater induction in response to As(V) compared to As(III). These patterns vary for SH1S21 and SH0S7 strains; although they present observable differences, these are not statistically significant. The BdlA protein is responsible for biofilm dispersion (27), while LuxS acts as a quorum sensing detection mediator (28). Finally, *ywqC* or *tkmA* gene is responsible for post-translational regulation during biofilm formation (29) and showed significant induction in both arsenic conditions regarding control in the three strains. Nonetheless, although there are observable differences between both arsenic conditions, these are not significant in any case.

## Scanning electron microscopy

SEM was used to observe the morphology of *Exiguobacterium* strains during biofilm formation in the presence and absence of As. This is one of the most commonly used methods to obtain images of the EPS and the characteristic grouping and shapes of these bacteria. The morphology of *Exiguobacterium* during biofilm formation is shown in Fig. 4, where overall, a greater formation of biofilms was observed in the presence of As(V) compared to As(III)-treated and untreated cells. In the case of strains SH31 and

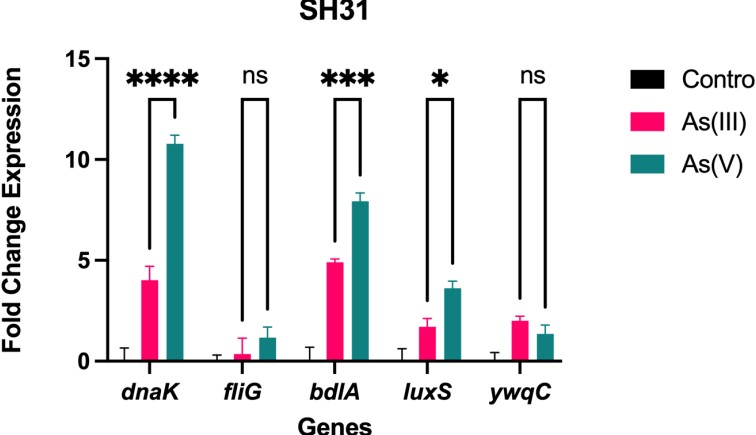

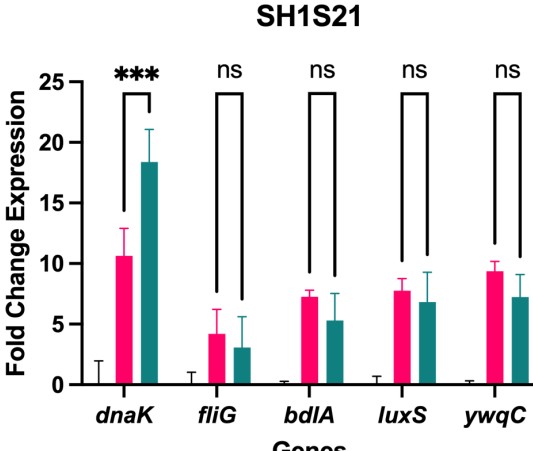

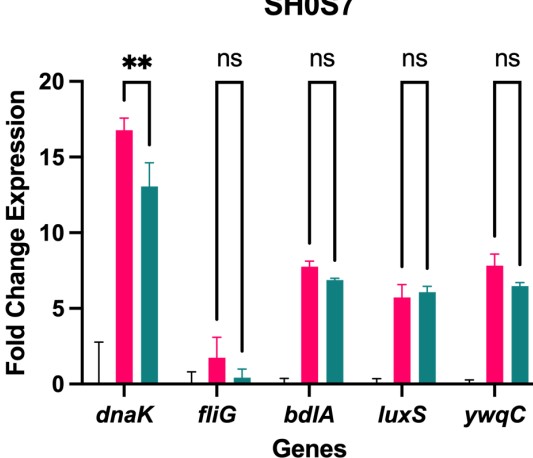

**FIG 3** Relative expression of genes related to biofilm formation of *Exiguobacterium* strains SH31, SH0S7, and SH1S21. The statistical analysis was performed by two-way ANOVA (ns: not significant, *$p < 0.05$, **$p < 0.01$, ***$p < 0.001$, ****$p < 0.0001$). Data show means of biological replicates ($n = 3$) with three technical replicates each ($n = 9 \pm$ SD).

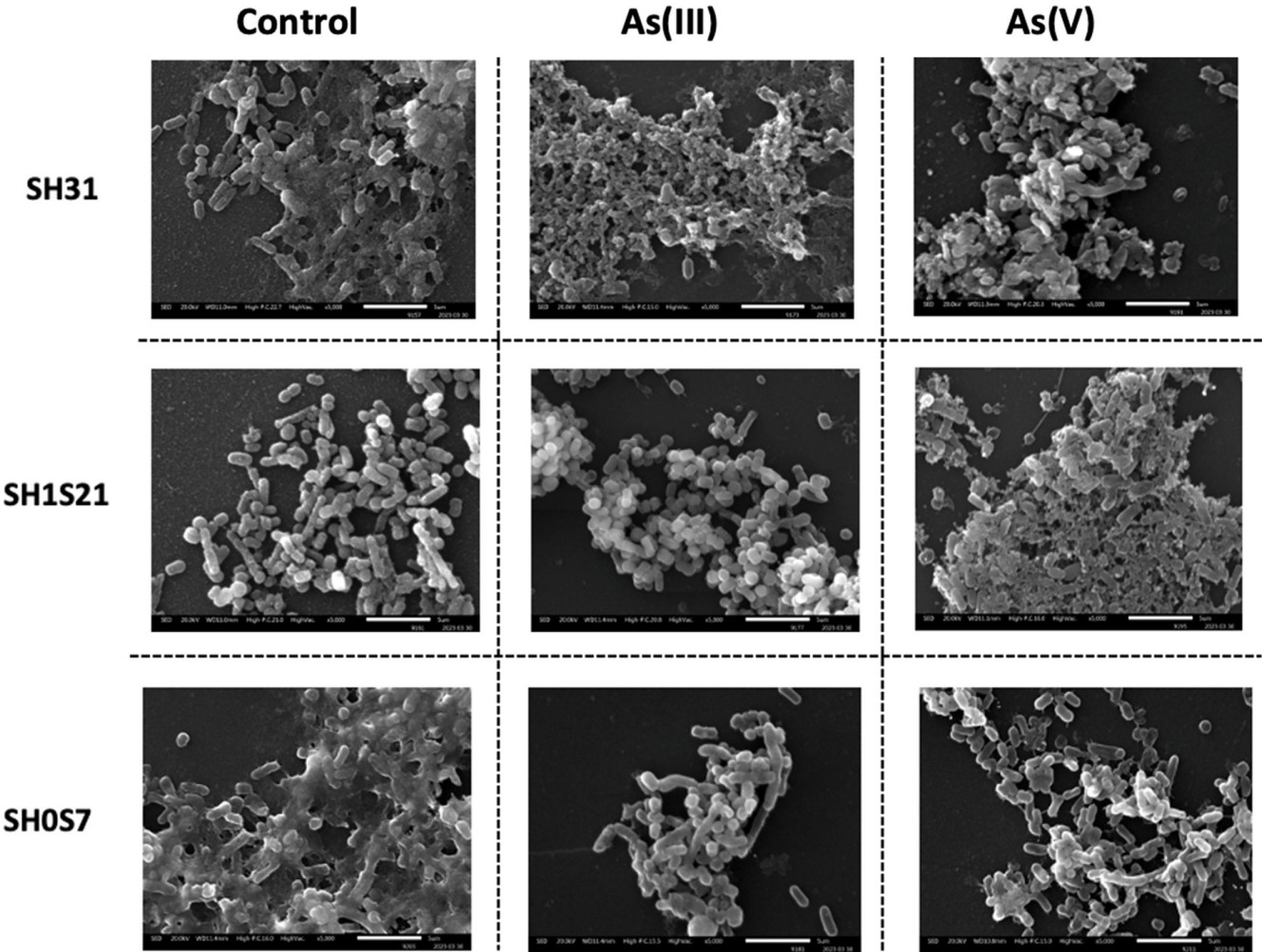

**FIG 4** Morphology of *Exiguobacterium* cells during biofilm formation. Conventional SEM images (5,000× magnification and the scale bar corresponds to 5 µm) in which grouped bacterial cells form biofilms in the presence and absence of As(III) and As(V). The concentrations used for each strain were SH31 (5 mM – 50 mM), SH1S21 (0.5 mM – 75 mM), and SH0S7 (10 mM – 75 mM), respectively.

SH1S21, greater adherence between bacteria was observed, while strain SH0S7 showed interbacterial adherence but without the appearance of EPS. When strain SH1S21 was treated with As(III), bacterial adherence was observed between those in a sessile state and free bacteria in a planktonic state; the latter also occurred in strains SH31 and SH0S7 incubated with As(III), with the difference that no biofilm formation was detected. Finally, all the bacteria treated with As(III) showed adherence properties, in that greater biofilm formation was observed, with more bacteria in a sessile state than in a planktonic state.

## Atomic force microscopy

To observe the structure of the biofilm produced by the *Exiguobacterium* strains, AFM was employed. The topography and 3D images observed in Fig. 5 are consistent with the findings of the CV staining assay (Fig. 2), in that the presence of As(V) triggered a greater formation of biofilm compared to the As(III)-treated and untreated samples of all three strains (SH31, SH1S21, and SH0S7). In Fig. 5, the intense yellow peaks indicated with an arrow correspond to the biofilm, whereas the darker areas highlight bacteria in a planktonic state, without biofilm formation, as depicted for strain SH1S21 in control conditions.

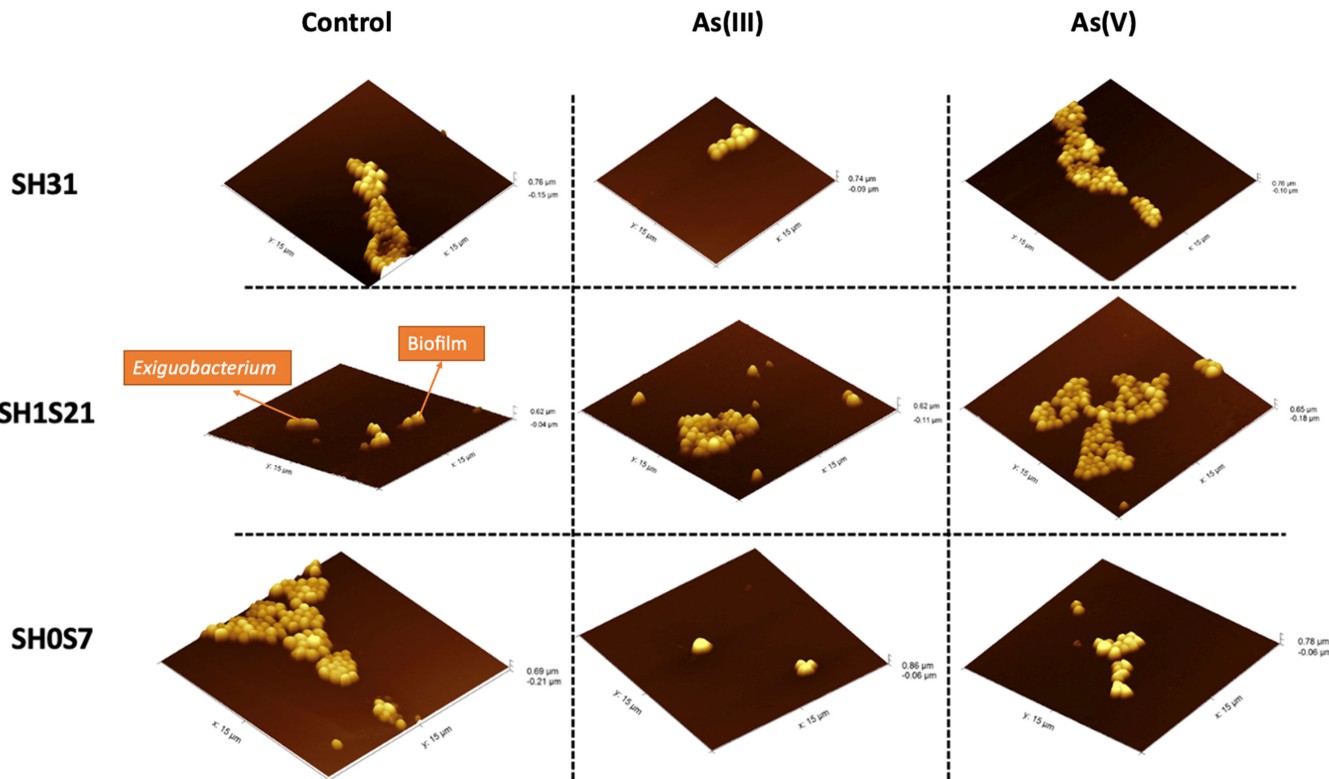

**FIG 5** Analysis of biofilm formation by *Exiguobacterium* strains using atomic force microscopy. The images correspond to the untreated strain (control +) and the strain treated with half of the As(III) and As(V) MIC.

## Composition of extracellular polymeric substances

An extraction of total EPS was performed to quantify and determine the composition of these extracellular elements in each of the strains studied, both in the presence and absence of As. Given that environmental bacteria produce more EPS when exposed to biotic or abiotic stress (30–32), EPS synthesized in SH strains were quantified at the end of the fermentation period (48 h) after exposure to As. All strains tested, SH31, SH1S21, and SH0S7, secreted higher levels of EPS (3.3 g/L, 2.8 g/L, and 2.75 g/L, respectively) when grown in the presence of As(V) compared to the amounts obtained in the presence of As(III) (1.8 g/L, 1.3 g/L, and 1.4 g/L, respectively). In particular, strain SH31 was the only strain that significantly increased EPS production (3.3 g/L) when exposed to As(V) compared to treatment without the toxin (2.2 g/L, 2.6 g/L and 2.5 g/L, respectively). For the case of strains SH1S21 and SH0S7, when exposed to As(III), EPS synthesis was significantly decreased compared to the untreated control (Fig. 6A). In addition, the neutral polysaccharide, protein, and DNA contents of the purified EPS were determined. High levels of sugars were found in the EPS synthesized by the strains analyzed, representing more than 70% of the total EPS, except for strain SH1S21. Protein and DNA contents were similar in all conditions evaluated, not exceeding 10% and 3%, respectively (Fig. 6B).

## Surface arsenic bio-adsorption

Fourier-Transform InfraRed (FTIR) spectroscopy was employed to determine whether bio-adsorption of As(III) and As(V) onto the surface of *Exiguobacterium* cells produced changes in functional groups that could be involved in metalloid binding. Effectively, changes were observed in the IR spectra of the three strains exposed to As (Fig. 7). In all cases, two main bands were observed (3,300 and 1,600 $cm^{-1}$), the first

**A**

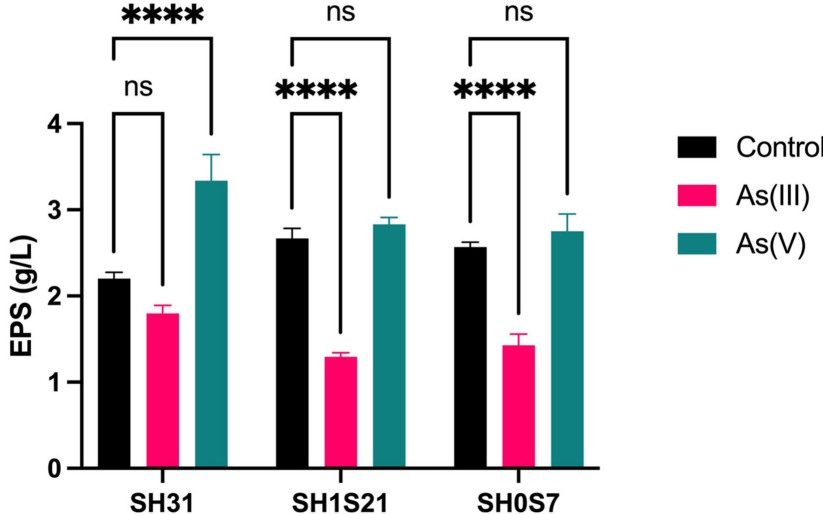

**B**

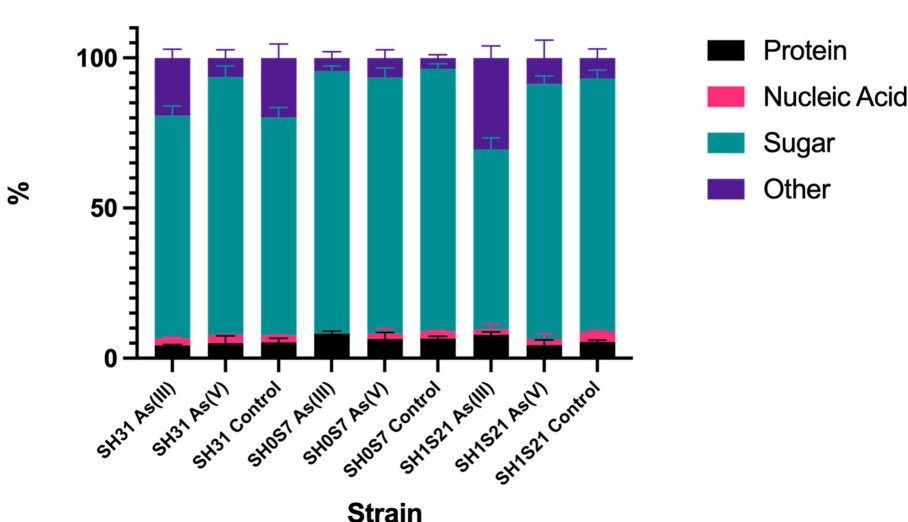

**FIG 6** Determination of the yield (A) and composition (B) of EPS synthesized by *Exiguobacterium* strains SH31, SH1S21, and SH0S7 in the presence and absence (control) of As(III) and As(V). The statistical analysis was performed by two-way ANOVA (ns: not significant, ****$p < 0.0001$). Bars represent means ($n = 3 \pm$ SD).

corresponding to OH-stretching and the second corresponding to the stretching of a C = C bond, according to a previous report (33). In addition, alterations were noted in the wavelengths of functional groups associated with specific biomolecules, as branched polysaccharides with the presence of OH-band intermolecular bonds (3,200–3,550 cm$^{-1}$), lipids (2,800–2,970 cm$^{-1}$), proteins (1,200–1,700 cm$^{-1}$) and simple polysaccharides (900–1,150 cm$^{-1}$), where the latter constitute the region known as the "fingerprint" (1,500–800 cm$^{-1}$) (34). Particularly, in the SH31 strain, the IR spectra were mostly equivalent in the case of both As species, unlike the control condition pattern where changes were detected, such as a notable increase in the fingerprint zone, specifically in peaks related

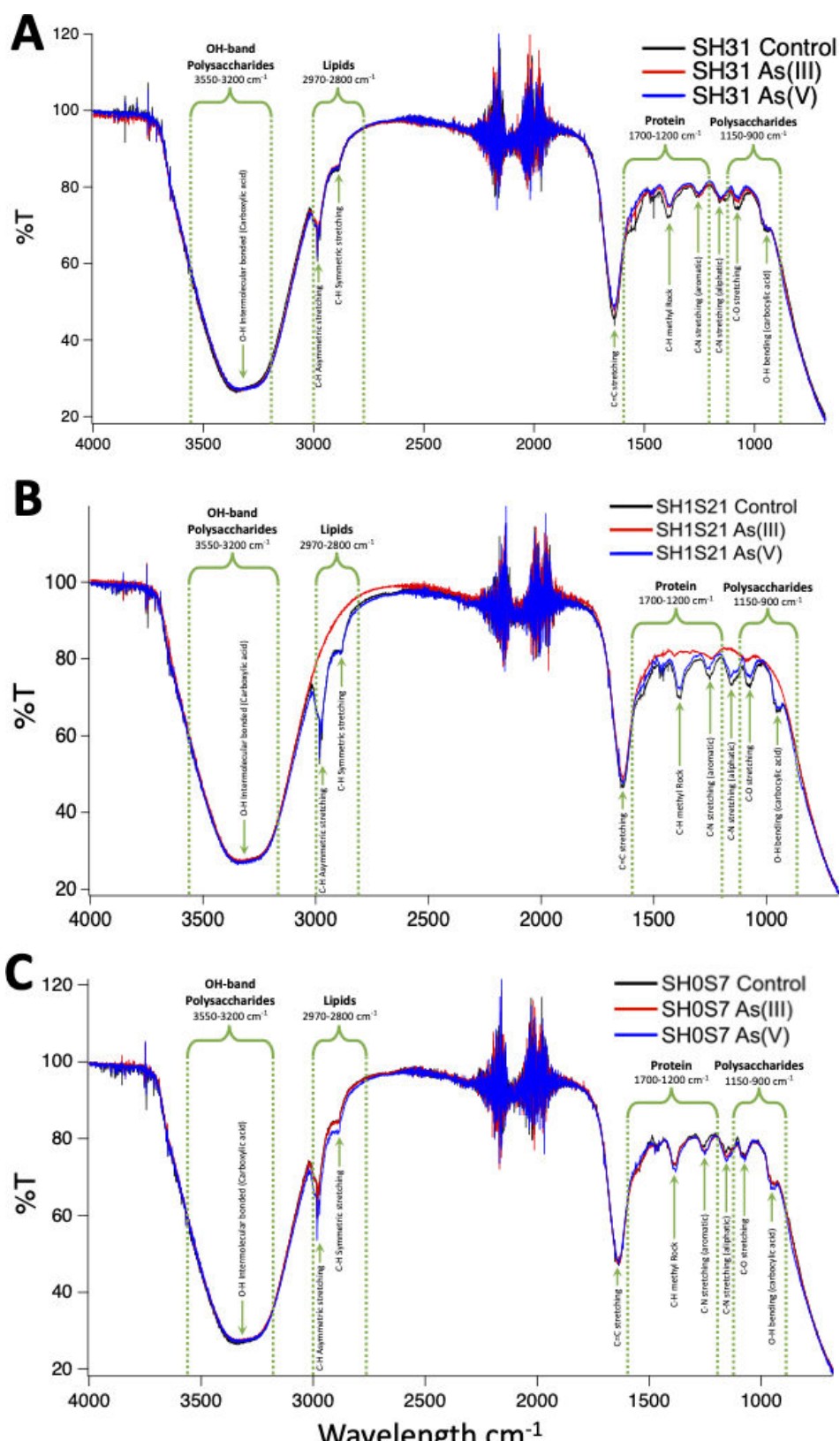

**FIG 7** FTIR spectra of As(III) and As(V) adsorption onto the cell surface of the *Exiguobacterium* sp. SH31 (A), SH1S21 (B), and SH0S7 (C) strains. The graph shows the relationship between transmittance (Y axis) and wavelength (X axis). The spectra colors correspond to: control (black), As(III) (red), and As(V) (blue).

to both proteins and polysaccharides. Moreover, the IR spectra of SH1S21 exposed to As(III) suffered an overall decline and the disappearance of the 3,000 and 2,800 cm$^{-1}$ peaks (C-H stretches) as well as all of the fingerprint region. On the other hand, a general increase compared to the control condition was observed in the As(V) treatment, in all peaks of the fingerprint region. Finally, the IR spectra of SH0S7 were differentiated by the presence of a slight increase in most of the peaks after exposure to As(V), compared to both control and As(III) conditions. These results suggest that several functional groups associated with different kinds of molecules could be involved in the adsorption of As species.

## DISCUSSION

The Salar de Huasco salt lake, like many landscapes in the Chilean highlands, is characterized by being an extreme yet highly variable environment, which makes it challenging for organisms to thrive in such conditions. Despite the harsh conditions, including large temperature variations, UV radiation, salinity, and the presence of toxic compounds such as As, many microorganisms inhabit this hostile place (35, 36). These microorganisms have become a study model that challenges researchers to understand how these bacteria can survive in an environment with high concentrations of heavy metals, including As. One of the proposed resistance mechanisms to As is the formation of biofilms in the *Exiguobacterium* strains SH31, SH1S21, and SH0S7 isolated from SH. Recent studies indicate that the SH31 strain, in the presence of high concentrations of NaCl, synthesizes more biofilm so that the bacteria can use it as a buffering resistance structure in saline conditions (37). This is common in extremophilic bacteria, as biofilms provide protection against extreme abiotic stress environments (38).

In this study, the aim was to evaluate the characteristics of biofilms as an adaptive mechanism that contributes to the resistance to As(III) and As(V) in strains isolated from the Salar de Huasco in northern Chile. First, the MIC of each strain against As(V) and As(III) was evaluated, revealing that the SH1S21 strain showed 10 times lower resistance to As(III) compared to the SH31 strain (1 mM and 10 mM, respectively). However, when the evaluated strains were exposed to As(V), all showed high resistance, with MIC values exceeding 100 mM, which significantly surpasses the resistance observed for As(III) (Table 3). These results are consistent with previous findings, suggesting that the Acr3 pump may be responsible for the increased resistance to As(III) and that the *ars* genes, clustered in an operon, contribute to As resistance, as found in several polyextremophiles (*arsRDABC*) (20, 21, 39).

A correlation was found between the local As concentrations at the sample sites from which the strains were isolated and their ability to resist As. The strains used in this study were isolated from different areas within SH, which have different concentrations of As (Table 1) (16). The predominant form of As in these places is expected to be arsenite, As(III), as it is generally found in alkaline environments with a pH above 7 (19, 40), as determined in all the sampled sites (pH 8–9). Therefore, it is expected that the strains found in these places activate detoxification processes from arsenate to arsenite (41). On the other hand, it has already been demonstrated that biofilm formation is a strategy used by bacteria to resist stressful and unfavorable conditions (30). Furthermore, under the studied conditions, biofilm formation was not affected by the presence of the metalloid; however, each strain showed a different growth pattern (Fig. 2). Nonetheless, As(V) induced a greater formation of biofilm, unlike the same strains exposed to As(III), where biofilm formation fell. This is consistent with previous studies that show that *Exiguobacterium* strains have higher expression of genes and proteins related to biofilm formation and response to As stress compared to the untreated strain (17, 37). Moreover, in a recent study, it was observed that the predominant resistance mechanism in *Exiguobacterium* strains is the reduction of As(V) to As(III), regulated by different genes, mainly by *acr3*, but also by *ars*B and *ars*J (20). Therefore, this reduction phenomenon forms one of the strategies that the tested *Exiguobacterium* strains possess to resist the presence of As in the environment.

To further analyze the biofilm and its structure, AFM was performed, which provides important information about the topography of the biofilm formed. For 48 h, the strains SH1S21, SH31, and SH0S7 were exposed to different conditions both in the absence and presence of As(III) and As(V). The images (Fig. 5) correlate with the data shown in Fig. 2 as they provide information about the differences in surface area and structure present in each treated strain. In these images, it was observed that the SH0S7 strain, when treated with As(III), exhibited greater height but with a lower number of peaks, which is an indicator of lower biofilm formation. In the case of the SH1S21 strain, bacteria were observed in the analyzed sample, but the formation or presence of a biofilm was not detected (arrow in Fig. 5), which typically occurs because there is no adherence state of the bacteria (42).

To complement the morphological analysis of *Exiguobacterium* and the consequent formation of biofilms, a SEM analysis of the strains in the presence of the toxin was performed. Previous reports indicate that the genus *Exiguobacterium* is morphologically characterized as a Gram-positive bacillus (15). In this context, when analyzing the images in Fig. 4, it was determined that the strains from SH, with and without As treatment, exhibited the characteristic rod-shaped form. Additionally, in Fig. 2 to 5, biofilm formation was observed in the presence of the metalloid species, and different stages of the biofilm could be identified, such as bacterial attachment, maturation, detachment, and dispersion (43). In the control images in Fig. 4, the EPS in the treated strains were clearly and distinctively visible. However, the SH31 and SH1S21 strains were adhesive, as in this phase, bacteria formed an EPS matrix, which is consistent with the results shown in Fig. 2. On the other hand, in SH0S7, bacteria were observed either recently adhering to each other or in the proliferation phase for subsequent detachment (Fig. 4). As for the SH0S7 strain, only bacteria that could be initiating biofilm formation were observed (44). Finally, regarding the treatment with As(V), the SEM images show that the SH1S21 and SH0S7 strains adhered through their EPS, in accordance with the results obtained in Fig. 2 and 5.

Based on the evidence obtained via AFM and SEM, we propose that biofilm formation is a critical strategy used by *Exiguobacterium* strains to resist As in the geologically rich Salar de Huasco, as both the formation of the biofilm and the phenomenon of bacterial adherence, promoted by EPS synthesis, suggest that the response of the tested strains is related to a defense and/or response process against this toxic compound present in their environment. The survival of *Exiguobacterium* in extreme environments is likely to be multifactorial, where multiple molecular mechanisms are activated to cope with the prevalent abiotic stresses. Biofilm formation, as described in this study, is one of several coordinated responses that these microorganisms can activate, opening a window for the investigation of the associated mechanisms that regulate its synthesis. Although the results clearly demonstrate that biofilms are the mechanism contributing to As resistance, the molecular processes contributing to the capacity of *Exiguobacterium* strains to tolerate the presence of toxic compounds, such as As, still need to be elucidated (45).

In order to contribute to the analysis of the regulatory mechanisms underlying biofilm formation in As treatments, the expression of key genes was analyzed in strains SH31, SH1S21, and SH0S7 exposed to the metalloid, particularly those involved in resisting the oxidative stress conditions generated by As (46). Prior to this study, it had been described in *Exiguobacterium* that the *dna*K gene responds to As-induced stress. This gene is a homolog of the Hsp70 chaperone, which promotes the biogenesis of curli, extracellular amyloids that are the main polymeric substances that modulate and colonize the biofilm by adhering to the surface and anchoring the cells (17, 47). In the SH31 strain, as shown in Fig. 3, the *dna*K gene was expressed 50% more under As(V) conditions compared to As(III); the expression of this gene in the SH1S21 strain responded in a similar manner. The SH0S7 strain expressed fivefold more *dna*K under As(III) than under As(V). In contrast, the expression of the *fli*G gene (which encodes for a flagellar protein) was lower in all three strains after exposure to As. This is because

once the bacteria adhere to the surface, this motility gene is inhibited. However, there is also evidence showing that the expression of genes related to biofilm formation inhibits motility (48), even though the flagellum allows bacteria to better adapt to environmental changes, avoiding harmful or stressful conditions (49).

It is necessary to consider that the last step in biofilm formation is dispersion. After leaving the matrix, bacteria can reattach to a surface, initiating the formation of a new biofilm or returning to a planktonic state. The *bdl*A gene is responsible for regulating chemotaxis, an essential phenomenon for biofilm dispersion (50–52). Consistently, in this context in the SH strains, there was 5 to 10 times higher expression of this gene in both As(III) and As(V) treatments compared to the control. Similarly, *lux*S also showed increased expression in the presence of the metalloid, a gene related to stress responses, quorum sensing, communication and detection, and biofilm formation. Quorum sensing is a bacterial cell-to-cell communication process where bacteria coordinate their gene expression by producing and releasing autoinducer molecules that allow them to adapt to unfavorable environmental conditions (53, 54). Finally, the *ywq*C or *tkm*A gene is a tyrosine kinase modulator that acts as an adjuvant to PTkA in phosphorylating proteins during post-translational regulation in biofilm formation. Available evidence shows that its absence leads to a loss in biofilm formation. In this study, the results indicate a higher expression of this gene in SH strains treated with As(III), but not As(V), possibly because the former species is more toxic (55).

The relative expression determination of genes associated with biofilm formation allows us to evaluate the biofilm formation process from a broader point of view. As mentioned before; *dnaK* gene is particularly important for the process, our results show that this gene responds to As(III) and As(V) by activating EPS biosynthesis in the tested strains, which are among the main catalysts for biofilm generation through cell adhesion and anchoring processes (Fig. 6). Also, EPS production and composition are fundamental signatures of biofilm formation, as these substances play a crucial role in the adherence of bacterial cells to the surface and in the three-dimensional structure of the biofilm.

Indeed, the results obtained in this study indicate that different strains of *Exiguobacterium* respond differently to the presence of As(III) and As(V) in terms of EPS production (Fig. 6). The SH31 strain showed higher EPS production in the presence of As(V), suggesting a specific response of this strain to this type of As. The increase in sugar content in the EPS of this strain could indicate a higher capacity for adherence and extracellular matrix formation, facilitating the formation of a robust biofilm, despite the IR spectra showing an increase in polysaccharide- and protein-related groups in the control condition (Fig. 7). On the other hand, the SH1S21 strain showed a decrease in EPS production in the presence of As(III), which may be related to a lower capacity for adherence and biofilm formation under these conditions, as indicated by the disappearance of peaks associated with these groups in the IR spectrum. The SH0S7 strain, although it follows the same pattern of a decrease in EPS in As(III), showed only a slight increase in protein content and the disappearance of nucleic acids in this condition, while the IR spectrum showed only a slight increase in peaks related to protein groups.

Our findings suggest that the response of *Exiguobacterium* strains to the presence of As is closely related to EPS production and composition. The ability of these bacteria to modulate EPS production depending on the type of As present in the environment may be an adaptive mechanism that allows them to survive and colonize contaminated niches. Taken together, the results obtained in the various assays performed on the studied strains show that exposure to As modulates the formation, structure, and composition of the generated biofilm. Clearly, each strain responds differently to the presence of the metalloid, which is likely related to its origin and the specific As content at a given location.

## Conclusions

In conclusion, the ability for biofilm formation by the *Exiguobacterium* strains examined in this study was not affected by the presence of the As metalloid, but each strain

exhibited a different growth pattern in response to different forms of the metalloid. Exposure to As(V) was found to induce greater biofilm formation, while exposure to As(III) resulted in lower biofilm formation. Furthermore, a correlation was identified between the sample sites and the resistance capacity of the isolated strains to As. Strains isolated from sediments with higher As concentrations showed higher resistance to As(V) and relatively lower resistance to As(III). The results highlight the importance of biofilm formation and the presence of specific resistance mechanisms in the ability of microorganisms to survive and thrive under adverse conditions.

## ACKNOWLEDGMENTS

We are very grateful for the help of Rocío Orellana from the Faculty of Dentistry, University of Chile, in obtaining the SEM images and Leonardo Caballero from the Department of Physics, University of Santiago, for the AFM analysis.

This research was sponsored by grants from ANID (Chilean National Research and Development Agency). C.P.S. was financed by ANID-FONDECYT Regular 1210633 and ANILLO-ANID ATE220007. F.R. was financed by ANID-FONDECYT Regular 1220902. J.C.-S. was financed by ANID 2021 Post-Doctoral FONDECYT 3210156. C.P.-E. was financed by ANID 2023 Post-Doctoral FONDECYT 3230189. G.K. was financed by ANID doctoral grant 21231337. K.G. was financed by ANID 2023 FONDECYT Iniciación 11230831. Also, this work was financed by the FONDEQUIP EQM130076 project to R.V. and by the FONDE-QUIP EQM180139 project to C.A. The sponsors and financing agencies had no role in the study design, data collection and analysis, the decision to publish, or the preparation of the manuscript.

C.P.S. conceived and designed the study. V.B.P., J.A., P.Z., G.K., and C.P.-E. performed all the biofilm formation and gene expression experiments. V.B.P., F.M., C.A., and R.V. performed all microscopy procedures. N.P. and V.B.P. conducted the EPS formation and analysis experiments. J.C.-S., F.R., and K.G. conducted the ATR-FTIR experiments and analyzed and interpreted the results. C.P.S., F.M., R.V., and F.R. contributed reagents, materials, and analysis. V.B.P., N.P., J.C.-S., C.P.-E., and C.P.S. analyzed and interpreted all the data and wrote the manuscript. All authors read and approved the final document.

The authors declare that the research was conducted in the absence of any commercial or financial relationships that could be construed as a potential conflict of interest.

## AUTHOR AFFILIATIONS

[1]Laboratorio de Microbiología Molecular, Facultad de Ciencias de la Vida, Universidad Andrés Bello, Santiago, Chile

[2]Laboratorio de Microbiología Aplicada y Extremófilos, Departamento de Ingeniería Química, Universidad Católica del Norte, Antofagasta, Chile

[3]Laboratorio de Ecología Molecular y Microbiología Aplicada, Departamento de Ciencias Farmacéuticas, Facultad de Ciencias, Universidad Católica del Norte, Antofagasta, Chile

[4]Laboratory of Allergic Inflammation, Department of Immunology and Microbiology, University of Copenhagen, Copenhagen, Denmark

[5]Centro de Investigación Tecnológica del Agua en el Desierto (CEITSAZA), Universidad Católica del Norte, Antofagasta, Chile

[6]Departamento de Química, Universidad Católica del Norte, Antofagasta, Chile

[7]Laboratorio de Física no Lineal, Departamento de Física, USACH, Santiago, Chile

[8]Laboratorio de Biología Periodontal, Facultad de Odontología, Universidad de Chile, Santiago, Chile

[9]Laboratorio de Microscopía Avanzada, Departamento de Ciencias Biológicas y Biodiversidad Universidad de Los Lagos, Osorno, Chile

## AUTHOR ORCIDs

Rolando Vernal  http://orcid.org/0000-0002-1391-320X
Claudia P. Saavedra  http://orcid.org/0000-0002-4248-8556

## FUNDING

| Funder | Grant(s) | Author(s) |
|---|---|---|
| ANID | Fondo Nacional de Desarrollo Científico y Tecnológico (FONDECYT) | 1210633 Regular | Claudia P. Saavedra |
| Agencia Nacional de Investigación y Desarrollo (ANID) | ATE220007 anillo | Claudia P. Saavedra |
| ANID | Fondo Nacional de Desarrollo Científico y Tecnológico (FONDECYT) | 1220902 Regular | Francisco Remonsellez |
| ANID | Fondo Nacional de Desarrollo Científico y Tecnológico (FONDECYT) | 3210156 PostDoc | Juan Pablo Castro-Severyn |
| ANID | Fondo Nacional de Desarrollo Científico y Tecnológico (FONDECYT) | 3230189 PostDoc | Coral Pardo-Esté |
| Agencia Nacional de Investigación y Desarrollo (ANID) | 21231337 Beca Doctorado | Gabriel Krüger |
| ANID | Fondo Nacional de Desarrollo Científico y Tecnológico (FONDECYT) | 11230831 Regular | Karem Gallardo |

## AUTHOR CONTRIBUTIONS

Valentina B. Pavez, Data curation, Formal analysis, Methodology, Writing – original draft | Nicolás Pacheco, Formal analysis, Methodology, Writing – original draft | Juan Castro-Severyn, Formal analysis, Investigation, Methodology | Coral Pardo-Esté, Methodology, Supervision, Validation | Javiera Álvarez, Investigation, Methodology | Phillippi Zepeda, Methodology | Gabriel Krüger, Methodology, Software, Validation | Karem Gallardo, Formal analysis, Methodology | Francisco Melo, Methodology, Supervision | Rolando Vernal, Methodology, Supervision | Carlos Aranda, Methodology | Francisco Remonsellez, Funding acquisition | Claudia P. Saavedra, Conceptualization, Formal analysis, Funding acquisition, Investigation, Project administration, Supervision, Writing – review and editing

## ADDITIONAL FILES

The following material is available online.

### Open Peer Review

**PEER REVIEW HISTORY (review-history.pdf).** An accounting of the reviewer comments and feedback.

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
