## [Reviewer comments · Microbiology Spectrum]

Microbiology Spectrum

Characterization of biofilm formation by *Exiguobacterium* strains in response to arsenic exposure.

Valentina Pavez, Nicolás Pacheco, Juan Castro-Severyn, Coral Pardo-Esté, Javiera Alvarez, Phillippi Zepeda, Gabriel Krüger, Karem Gallardo, Francisco Melo, Rolando Vernal, Carlos Aranda, Francisco Remonsellez, and Claudia Saavedra

Corresponding Author(s): Claudia Saavedra, Universidad Andres Bello

Review Timeline:

Submission Date:	June 27, 2023
Editorial Decision:	July 22, 2023
Revision Received:	July 31, 2023
Editorial Decision:	August 14, 2023
Revision Received:	August 16, 2023
Accepted:	August 19, 2023

Editor: Silvia Cardona

Reviewer(s): The reviewers have opted to remain anonymous.

Transaction Report:

DOI: <https://doi.org/10.1128/spectrum.02657-23>

July 22, 2023

Dr. Claudia P. Saavedra
Universidad Andres Bello
Lab. Microbiología Molecular, Departamento de Ciencias Biológicas
Republica 330
Santiago, Santiago 8370186
Chile

Re: Spectrum02657-23 (Characterization of biofilm formation by *Exiguobacterium* strains in response to arsenic exposure.)

Dear Dr. Claudia P. Saavedra:

Thank you for submitting your manuscript to Microbiology Spectrum. Your article was evaluated by two experts in the field. Both reviewers agree that your manuscript is a well-organized and interesting report that contributes to knowledge of the topic. In addition, reviewers have raised some points, that need to be addressed before the article can be accepted for publication. Importantly, one reviewer pointed out that the statistical analysis needs to be revised and that conclusions should be made according to statistically significant results. Please, pay close attention to this comment when working on the revised version of your article.

When submitting the revised version of your paper, please provide (1) point-by-point responses to the issues raised by the reviewers as file type "Response to Reviewers," not in your cover letter, and (2) a PDF file that indicates the changes from the original submission (by highlighting or underlining the changes) as file type "Marked Up Manuscript - For Review Only". Please use this link to submit your revised manuscript - we strongly recommend that you submit your paper within the next 60 days or contact me. Detailed instructions on submitting your revised paper are below.

Link Not Available

Saludos!

Silvia Cardona

Journals Department
Reviewer comments:

Reviewer #1 (Comments for the Author):

Line No. Comment

2 Remove full stop from the heading
47 Change to enhanced
174 Change to samples
224 Change to 10ml
233 Remove then
327 Use symbol of micro not the letter "u"
349 Remove full stop from the subheading
403 Replace word demanding with difficult or
challenging
484 Change "are an event" with better word(s)
500 Remove "in contrast"

The Conclusions section should be more precise and brief. Remove the points already discussed in the Introduction and Results section.

There should be consistency in using the verb tense as at places it is in present where as at others it is in past. Better to get the manuscript reviewed by the person proficient in English.

Reviewer #2 (Comments for the Author):

Review of "Characterization of biofilm formation by *Exiguobacterium* 1 strains in response to arsenic exposure" by Pavez et al.

The authors have submitted their analysis of several strains of *Exiguobacterium* in response to treatment with As(III) and As(V). These strains were isolated from extreme environments with high exposure to As. They focus in particular on the response of biofilms formed by the bacteria, and some genes associated with this process. They show that biofilm formation seems to contribute to enhancing the arsenic resistance of these strains. This is an interesting report, and the data are important as understanding As resistance is important for many reasons, including development of bioremediation processes. I have several questions for the authors.

Line 116 - Please define H0, H1, and H4 in the figure legend, as well as the yellow spot, which I assume is the location of the lake.

L 162 - please give a reference (or the recipe) for LB + NaCl medium

L 281 - The figure legend is a bit confusing. The authors say, "...concentrations used for each strain were SH31 (5 mM - 50 mM)...". Does this mean a range of concentrations? Or do you mean the concentrations were 5 mM for As(III) and 50 mM for As(V) respectively?

L 294 - Is there a statistically significant difference in DnaK expression in SH0S7? They appear to be pretty much the same, not higher in As(III) by eye.

L 294- Same question for bdlA. You don't show any p value data for the two conditions, only for each condition against control. Expression of bdlA doesn't appear to be different in the two conditions on AsIII, yet you seem to conclude there is a significant difference.

Same question for luxS and ywqC - by eye there is no appreciable difference in expression between AsIII and AsV.

Line 355 - you state all three strains produce more EPS in response to AsV relative to control, but the data in Figure 6 are not significant for two of the strains.

Line 359 - Exposure to AsIII did not reduce EPS for strain SH31. I think you need to be clear where there were statistically significant changes, and where there were not in the text.

Minor points:

Line 646 - Reference 10 needs a title

Figure 7 - could you make the legend lines (showing which color is for which treatment) thicker? It is difficult to tell which is which even when enlarged.

Staff Comments:

Preparing Revision Guidelines

Please return the manuscript within 60 days; if you cannot complete the modification within this time period, please contact me. If you do not wish to modify the manuscript and prefer to submit it to another journal, please notify me of your decision immediately so that the manuscript may be formally withdrawn from consideration by Microbiology Spectrum.

We thank the reviewers for their constructive comments that have improved the manuscript quality. We included all recommendations and present a point-by-point response to each concern and a marked-up version of the manuscript indicating all changes.

Reviewer comments:

Reviewer #1 (Comments for the Author):

Line No. Comment

2 Remove full stop from the heading

Done. Line 2.

47 Change to enhanced

Done. Line 48

174 Change to samples

Done. Line 185

224 Change to 10ml

Done. Line 237

233 Remove then

Done. Deleted Line 246

327 Use symbol of micro not the letter "u"

Done. Line 844

349 Remove full stop from the subheading

Done. Line 329

403 Replace word demanding with difficult or challenging.

Done. Line 375 changed to "challenging"

484 Change "are an event" with better word(s)

Done. Line 458 changed to "are the mechanism"

500 Remove "in contrast"

Done. Line 471

The Conclusions section should be more precise and brief. Remove the points already discussed in the Introduction and Results section.

The conclusion section was edited, and it is now more precise and improved overall. Lines 527-536

There should be consistency in using the verb tense as at places it is in present where as at others it is in past. Better to get the manuscript reviewed by the person proficient in English.

The whole manuscript was reviewed by a English native speaker

Reviewer #2 (Comments for the Author):

Review of "Characterization of biofilm formation by Exiguobacterium 1 strains in response to arsenic exposure" by Pavez et al.

The authors have submitted their analysis of several strains of Exiguobacterium in response to treatment with As(III) and As(V). These strains were isolated from extreme environments with high exposure to As. They focus in particular on the response of biofilms formed by the bacteria, and some genes associated with this

process. They show that biofilm formation seems to contribute to enhancing the arsenic resistance of these strains. This is an interesting report, and the data are important as understanding As resistance is important for many reasons, including development of bioremediation processes. I have several questions for the authors.

Line 116 - Please define H0, H1, and H4 in the figure legend, as well as the yellow spot, which I assume is the location of the lake.

Done. Definition added in the legend. Line 826-828.

L 162 - please give a reference (or the recipe) for LB + NaCl medium

Done. Added recipe. Line 172-174

L 281 - The figure legend is a bit confusing. The authors say, "...concentrations used for each strain were SH31 (5 mM - 50 mM)...". Does this mean a range of concentrations? Or do you mean the concentrations were 5 mM for As(III) and 50 mM for As(V) respectively?

Done. Yes, I was respectively but it was changed to be more clear. Line 832-834

L 294 - Is there a statistically significant difference in DnaK expression in SH0S7? They appear to be pretty much the same, not higher in As(III) by eye.

Done. The statistic was changed to be between arsenic. Line 292-294

L 294- Same question for bdlA. You don't show any p value data for the two conditions, only for each condition against control. Expression of bdlA doesn't appear to be different in the two conditions on AsIII, yet you seem to conclude there is a significant difference.

Done. The statistic was changed to be between arsenic. Figure 3.

Same question for luxS and ywqC - by eye there is no appreciable difference in expression between AsIII and AsV.

Done. The statistic was changed to be between arsenic. Figure 3.

Line 355 - you state all three strains produce more EPS in response to AsV relative to control, but the data in Figure 6 are not significant for two of the strains.

Done. The paragraph was modified. Line 337-342

Line 359 - Exposure to AsIII did not reduce EPS for strain SH31. I think you need to be clear where there were statistically significant changes, and where there were not in the text.

Done. The paragraph was modified. Line 337-342

Minor points:

Line 646 - Reference 10 needs a title

Done. Reference changed. Line 608-610

Figure 7 - could you make the legend lines (showing which color is for which treatment) thicker? It is difficult to tell which is which even when enlarged.

Done. The lines were modified and added in the legend of the image to which each color corresponds. Line 860-861

August 14, 2023

Dr. Claudia P. Saavedra
Universidad Andres Bello
Lab. Microbiología Molecular, Departamento de Ciencias Biológicas
Republica 330
Santiago, Santiago 8370186
Chile

Re: Spectrum02657-23R1 (Characterization of biofilm formation by *Exiguobacterium* strains in response to arsenic exposure.)

Dear Dr. Claudia P. Saavedra:

Thank you for submitting your manuscript to Microbiology Spectrum. As you will see your paper is very close to acceptance. However there is one outstanding issue raised by reviewers that need to be addressed. Please modify the manuscript along the lines reviewer 2 recommended. As these revisions are quite minor, I expect that you should be able to turn in the revised paper in less than 30 days, if not sooner. If your manuscript was reviewed, you will find the reviewers' comments below.

When submitting the revised version of your paper, please provide (1) point-by-point responses to the issues raised by the reviewers as file type "Response to Reviewers," not in your cover letter, and (2) a PDF file that indicates the changes from the original submission (by highlighting or underlining the changes) as file type "Marked Up Manuscript - For Review Only". Please use this link to submit your revised manuscript. Detailed instructions on submitting your revised paper are below.

Link Not Available

Sincerely,

Silvia Cardona

Reviewer comments:

Reviewer #1 (Comments for the Author):

Suggestions incorporated

Reviewer #2 (Comments for the Author):

Review of "Characterization of biofilm formation by *Exiguobacterium* strains in response to arsenic exposure" by Pavez et al. second submission

The authors have responded sufficiently to the previous reviews I think except for one point:

Line 298-303 and 497 - I still feel that the authors need to indicate, in the discussion here, that while the bars do seem to show (by eye) that, for example, ywqC is more highly expressed under As(III) conditions, the differences are not statistically significant, and thus can't really be said to be higher.

Preparing Revision Guidelines

Please return the manuscript within 60 days; if you cannot complete the modification within this time period, please contact me. If you do not wish to modify the manuscript and prefer to submit it to another journal, please notify me of your decision immediately so that the manuscript may be formally withdrawn from consideration by Microbiology Spectrum.

We thank the reviewer for their constructive comments that have improved the manuscript quality. We included all recommendations and present a point-by-point response to each concern and a marked-up version of the manuscript indicating all changes.

Reviewer comments:

Reviewer #2 (Comments for the Author):

Review of "Characterization of biofilm formation by *Exiguobacterium* strains in response to arsenic exposure" by Pavez et al. second submission

The authors have responded sufficiently to the previous reviews I think except for one point:

Line 298-303 and 497 - I still feel that the authors need to indicate, in the discussion here, that while the bars do seem to show (by eye) that, for example, *ywqC* is more highly expressed under As(III) conditions, the differences are not statistically significant, and thus can't really be said to be higher.

Done. The text was modified in two paragraphs to respond to the reviewer on the results and on the discussion.

Results Lines 276-294 "To better understand biofilm formation, the relative expression of five genes related to this process (stress response, motility, formation) was measured. In general, an induction of all evaluated genes was observed in response to As(III) and As(V) regarding control condition (Figure 3). While no significant differences were detected in most cases when comparing the patterns between As(V) and As(III). Of the mentioned genes, *fliG*, which codes for a protein that favors bacterial motility (24), displayed the lowest induction for all three strains (SH31, SH1S21 and SH31). On the other hand, *dnaK*, which plays an important role in the folding of proteins necessary for biofilm formation (25), shows higher expression for the strains SH31 and SH1S21 in response to As(V) than to As(III), which is opposed to what was observed in the SH0S7 strain. Moreover, *bdIA*, and *luxS* were the only two other genes that presented significant differences, only in the SH31 strain and in both cases with a greater induction in response to As(V) compared to As(III). These patterns vary for the SH1S21 and SH0S7 strains, although they present observable differences, these are not statistically significant. The BdIA protein is responsible for

biofilm dispersion (26), while LuxS acts as a quorum sensing detection mediator (27). Finally, the *ywqC* or *tkmA* gene is responsible for post-translational regulation during biofilm formation (28), showed a significant induction in both arsenic conditions regarding control in the three strains. Nonetheless, although there are observable differences between both arsenic conditions, these are not significant in any case”

Discussion lines 488-495 “The relative expression determination of genes associated with biofilm formation allow us to evaluate the biofilm formation process from a broader point of view. As mentioned before, *dnaK* gene is particularly important for the process, our results show that this gene responds to As(III) and As(V) by activating EPS biosynthesis in the tested strains, which are among the main catalysts for biofilm generation through cell adhesion and anchoring processes (Figure 6). Also, EPS production and composition are fundamental signatures of biofilm formation, as these substances play a crucial role in the adherence of bacterial cells to the surface and in the three-dimensional structure of the biofilm”.

August 19, 2023

Dr. Claudia P. Saavedra
Universidad Andres Bello
Lab. Microbiología Molecular, Departamento de Ciencias Biológicas
Republica 330
Santiago, Santiago 8370186
Chile

Re: Spectrum02657-23R2 (Characterization of biofilm formation by *Exiguobacterium* strains in response to arsenic exposure.)

Dear Dr. Claudia P. Saavedra:

Thank you for submitting your revised version. I am happy to let you now that your manuscript has been accepted, and I am forwarding it to the ASM Journals Department for publication. You will be notified when your proofs are ready to be viewed.

Best regards.

Sincerely,

Silvia Cardona
Editor, Microbiology Spectrum
